# Learning Performance Styles in Gamified College Classes Using Data Clustering

**Sungjin Park**  **and Sangkyun Kim ***

Graduate School of Business, Kyung Hee University, Seoul 02447, Republic of Korea
* Correspondence: saviour@khu.ac.kr; Tel.: +82-02-961-0127

**Abstract:** This study aimed to investigate the efficacy of learning gamification in developing sustainable educational environments. To this end, gamified class data were analyzed to identify students' learning performance patterns. The study sample comprised 369 data points collected across four point domains: Activity, Game, Project, and Exam Points, which students obtained in their gamified college courses conducted between 2016 and 2019. A K-means data clustering algorithm and silhouette analysis were utilized to evaluate student performances and determine differential learning styles in gamified environments. Cluster analysis revealed three types of learning patterns centered on performance, mastery, and avoidance. Based on our findings, we propose suggestions regarding class design for instructors considering using gamification strategies to support a sustainable educational environment. We also highlight the scope for future research in both in-person and online gamified learning.

**Keywords:** gamification; gameful experience; data clustering; K-means; silhouette coefficient

## 1. Introduction

Gamification refers to the application of game elements (e.g., points, badges, and a leaderboard) to non-game contexts such as management, education, and marketing, to deliver gameful experiences to users and increase user engagement and motivation [1]. Since late 2019, COVID-19 has affected communities globally with a significant impact on health and quality of life, including imposed restrictions on daily lives. Accordingly, education was transitioned to a full-scale distanced model. Gamification in education [2] has demonstrated the ability to ensure an uninterrupted and seamless learning experience amid the restrictions induced by COVID-19 [3].

As the educational effects of gamification have been increasingly recognized, the scope of its application has expanded further. Research on gamification design, particularly those concerning the metaverse, is in progress. According to Park and Kim [4], gamification is utilized in the metaverse to improve user participation and engagement. They suggested five world types to deliver gameful experiences in the virtual world metaverse.

As discussed above, the gameful experience has already been positioned as an approach to secure sustainability for continuous learning environment settings. Businesses are striving to design gameful experiences to market products and services to a greater number of users, and individuals and organizations use it as a medium to deliver value in specific contexts.

However, poor design may deprive users of gameful experiences or fail to influence them to change their behaviors [5]. Accordingly, it is necessary to conduct research on systematic gameful experience design for populations of all age ranges [6].

Many instructors consider employing a gameful experience design while planning sustainable education strategies for college students. Instructors consider components such as game elements, game rules, and interaction among learners. In gamified learning

environments, learners seek preferred play styles, play elements, and game rules [6], just as the 16 different MBTI personality types are expected to quest for different things.

This study aimed to examine the ways of improving gamified learning performances by identifying the various learning patterns in a gamified class and the essential components required for designing a gameful experience for each learning pattern. Our systematic approach was based on the following research question:

**RQ1.** *How many patterns of learning performance can be identified in a gamified class, and what are their characteristics?*

## 2. Materials and Methods

### 2.1. Gamified Class Design

The study was conducted using data obtained from college classes taught by members of the research team between the first semester of 2016 and the second semester of 2019 (Figure 1). The research team members taught a total of five courses covering the following three subjects: "Gamification Engineering," which taught gamification; "Entrepreneurship," which covered the definition and elements of entrepreneurship; and "Technology Innovation," which taught about the evolution of technology, need for innovation, and classification of technology innovations.

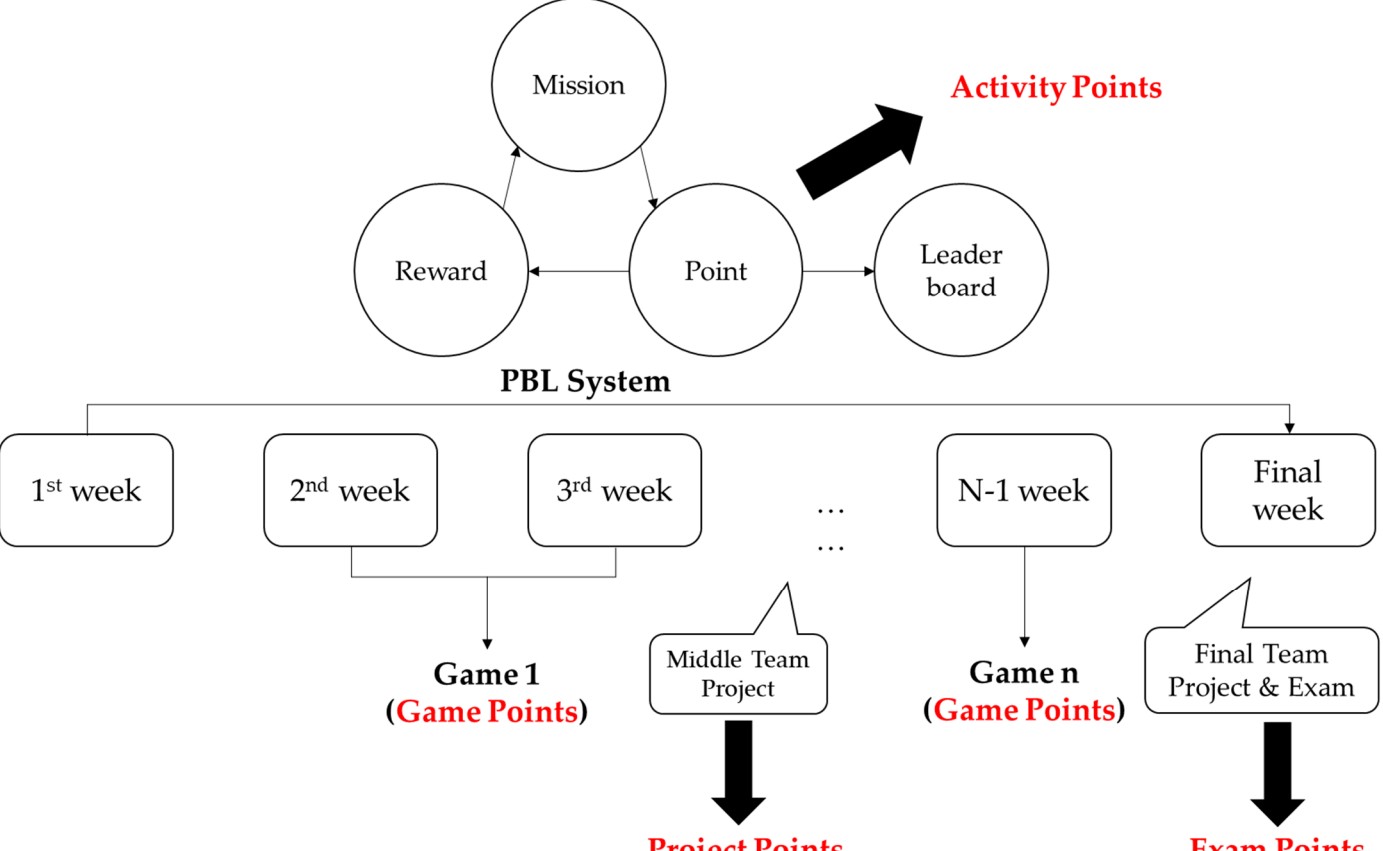

**Figure 1.** Flowchart of gamified coursework.

First, weekly syllabi were developed for each course. Then, gameful experiences were designed. Two gamification types were identified and employed in the present study by reviewing previous studies. The first was a points–badges–leaderboards (PBL) system and the second was a game play system.

For the flow chart above, the PBL system was applied to evaluate the learning activities of the students described in Section 2.1.1. The description of the game played from the 1st

week to the last week is in Section 2.1.2. Red words in the flow chart are the data collected for this study. The data collection process is described in Section 2.2.

### 2.1.1. PBL System Design

PBL is a gamified system that utilizes points, badges, and a leaderboard [7]. Instructors use this system to give learners missions to promote learning. Missions are determined based on difficulty as well as learner levels. Learners who accomplish a mission earn points, and they can use the points in exchange for rewards to support learning. Additionally, the total points each learner earns are visualized in a leaderboard to provide feedback to other learners. The PBL system is a basic element of gamification in education [3].

In our study, missions were announced as part of the syllabus given to all students. In addition, real-time missions were announced by the instructor during classes. The missions were designed to support learning and included asking questions in class, answering questions posed by the instructor to the entire class, and presenting learning activity sheets before the end of a class. To reward learners for accomplishing a mission, 1–3 points were given. For point management and leaderboard use, Class Management Platform Service (Figure 2), a service that provides mobile and web class support, was used. After either downloading the app or registering on the service's webpage, learners were able to check the points they and other learners had earned during the course by entering the class code provided by the instructor.

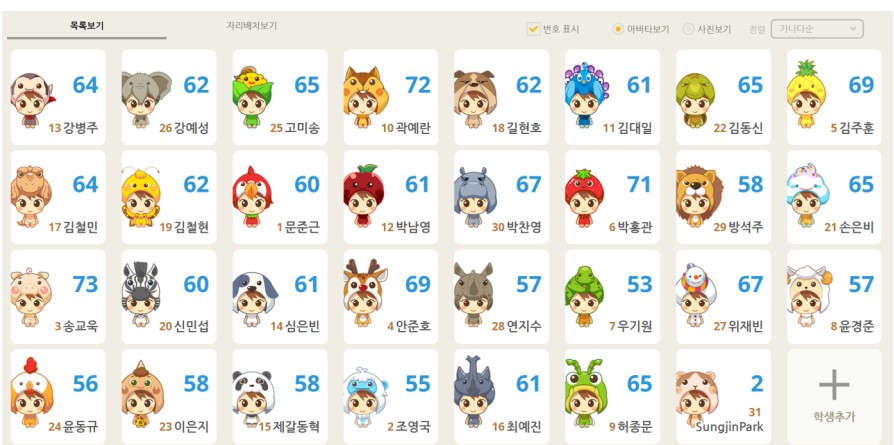

**Figure 2.** Leaderboard allowing students to check points earned (Korean text is student's name).

Rewards that could be exchanged for earned points were specified on magic cards (Figure 3). The magic cards were designed to serve as badges in authorization game mechanics. The number of exchangeable magic cards was limited, and point thresholds were set to ensure a learner earned a sufficient number of points before requesting an exchange. The cards used in exchange for other rewards could not be reused. Moreover, learners could not keep more than one card of the same type.

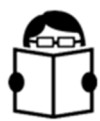

**Open Book Exam**

- Student can refer to their textbook for 1 minute during the exam.
- Present the card to the professor, and the professor will check the time.
- Do not show the textbook to other students.

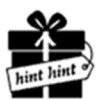

**Ultimate Clue**

- Student can ask the professor for a clue to a question in an exam.
- Present the card to the professor, and the professor will give you a clue.
- Do not share the clue with other students.

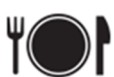

**Lunch with SJ Buffet**

- Student can have lunch with the professor and bring a friend of their choice.
- Present the card and schedule a time.
- The professor will buy a delicious meal.
- The menu will be determined by all parties.

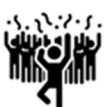

**Best Mercenary**

- Student can use an hour of the professor's time while doing a team project or studying for an exam.
- The professor will not participate in the activity can help by providing support or conveying important know-how.

**Figure 3.** Magic Cards obtained by earning points.

### 2.1.2. Game Design

The second type of gamification identified in the literature review was simulation game play of the class contents. Kim [8] developed a Salary Auction Game to enable learners to think about teamwork and salary and demonstrated that the game served as an effective educational tool to improve learning attitudes while developing a fun learning environment. In a study conducted with adults, Sidhu et al. [9] derived 10 elements involved in teaching entrepreneurship to adult learners and suggested the application of gamification to effectively deliver a lecture on each element.

Course contents were developed using board games, PC-based games (Web or software), and alternative reality games. The following games were applied to all courses in this study:

- Investment Game: In this game, learners established virtual companies in class or used the platform as a tool for peer evaluation of project outcomes. In the game, learners were provided with virtual currency, "Garnet," which they invested in companies established by other teams (but not by their own teams). The points they earned differed according to the return on investment, and players with higher returns received higher points [10];
- Devil's Advocate Game: This was a team activity game. Two members were recruited from each team and joined other teams in turn for a predetermined length of time to discuss into the disadvantages or flaws in the other teams' ideas and provide negative feedback. Points were earned if the negative feedback was rational;
- Salary Auction Game: This game was played in the beginning of each semester when the teams were formed. The purpose of this game was to minimize the learners' reluctance to participate in team activities and ensure that learners were mixed well [8].

In addition to the three games mentioned above, other games were developed in each course to deliver course-specific value [11]. At the end of each game, points were given according to the participants' performance, and the final scores reflected their earned points.

### 2.2. Grade Evaluation and Data Collection

To grade academic performance, four grade points, namely Activity, Game, Project, and Exam Points, were determined as follows:

- Activity Points: Points earned in the PBL system mentioned in Section 2.1.1. Points were earned when leaners accomplished a mission given by the instructor;
- Game Points: Points earned in the games mentioned in Section 2.1.2. Points were considered the sum of points earned in each game;

- Project Points: Points earned in two team projects (mid-term and final). Points were determined by adding the points given by the instructors to individual learners based on their project evaluation to the points earned by the team in the Investment Game;
- Exam Points: Points earned by each learner on tests throughout the course.

Grade points in each category were calculated using the following Formula (1):

$$\text{Grade point} = \frac{a\ student's\ points}{top\ player's\ points} \times the\ points\ assigned\ to\ the\ category \qquad (1)$$

For example, an instructor planned to distribute a total of 100 points across the four point categories to determine individual learners' academic performance in Course A, offered in the first semester of 2016. According to the institutional rule that the final exam score should have a weight of 20% or more, the instructor designed an academic performance evaluation system in which 20 out of the total of 100 points were assigned to Exam Points and the remaining 80 points were distributed across Activity (20 points), Game (30), and Project Points (30).

Activity Point scores were determined as follows. For example, in Course A, a total of 20 points were assigned to Activity Points. If Student B earned the most points at the end of the class (80 points), the student's Activity Point score was 20 according to Formula (1). Student C, who earned 60 points, received a score of 15 based on the same formula. Thus, all students received points according to their own performances in relation to the points earned by the highest scoring student.

Game Point scores were determined as follows. For example, in Course A, a total of 30 points were assigned to Game Points. If at the beginning of a semester, the instructor elected to assign 10 out of the 30 points to the Investment Game and Student B earned the most Garnets (150), this student was given a score of 10 according to the formula above. If Student C earned 140 Garnets, they received a score of 9.3. In this manner, all students were scored differentially in Game Point. Grades were determined based on points earned in all games.

Project Point scores were determined by first calculating the proportions of the score given by the instructor and adding it to the points invested by learners on other teams in the Investment Game. For example, in Course A in the first semester of 2016, a total of 30 points were assigned to Project Points. Assume that 10 of the 30 points were assigned to the mid-term project and 20 to the final project and that, in each project, 60% of a student's score was based on instructor evaluation and 40% on the number of points accumulated for their performance in the Investment Game. If, in the final project, Team B received 8 out of 12 points from the instructor evaluation and earned the most points in the Investment Game (400 Garnets), the team received a total score of 16 according to the formula above, because applying the formula to the Investment Game yielded 8 points (i.e., 40% of 20 points).

Exam Points reflected individual learners' exam scores. The number of questions in an exam differed across courses, but the total score was set to 20 regardless of the course. For instance, there were a total of 25 questions in the final exam of Course A taught in the 1st semester of 2016, 10 of which were easy and 15 were difficult. Easy questions were worth 0.5 points and difficult questions were worth 1 point, yielding a total possible score of 20 points.

Data for each of the four categories were collected as described in Table 1. Considering the differences in the number of students and activities per course, the data were standardized per semester. Specifically, standardization was performed for each category per semester to transform raw scores onto a scale of 0–1.

**Table 1.** Description of data collected for all courses.

| Year | Semester | Lecture | # of Data |
|------|----------|---------|-----------|
| 2016 | 2nd | Gamification Engineering | 247 |
| 2017 | 1st | Entrepreneurship | 29 |
|      | 2nd | Gamification Engineering | 30 |
| 2018 | 1st | Technology Innovation | 33 |
| 2019 | 1st | Technology Innovation | 30 |
|      |     | Total | 369 |

### 2.3. Data Clustering Algorithm

Data clustering is an unsupervised learning algorithm in which the computer automatically classifies data based on certain rules once a human specifies the analytical criteria. In the past, machine learning techniques employing classification algorithms aimed at classifying data according to user-defined criteria, whereas data clustering can help form sensible groups of data based on criteria not specified by the user, such as the use of data images and language interpretation [12]. For example, finance companies develop criteria to classify members according to credit level and label those who correspond to certain criteria. Even in subsequent data analysis, the labeled data are classified based on the user-specified criteria and the labels. In contrast, data clustering is an algorithm that labels previously unlabeled data based only on user-specified criteria and the analytical algorithm.

In this study, K-means was used as the clustering algorithm. The algorithm was first reported in 1955 and is the most frequently used, well-known, and simple clustering algorithm [12]. In this algorithm, data points close to a mean value are searched for and grouped together, and the clustering level is assessed using the Euclidean distance. The K-means algorithm works in the following way:

1. Setting parameters: Using the data in this study as an example, the number of data points is 369 and the number of attributes (i.e., the four categories mentioned above) is 4. Hence, a 369 × 4 two-dimensional matrix was created. Each data point (1st–369th data point) corresponds to the cluster in which it belongs. In addition, the appropriate number of clusters for the dataset should be specified. Thus, the number of clusters, k, should be specified and a solution display should be created to show the clusters and constituent data points. Different researchers used different methods to display a solution. For instance, Researcher A may have created a matrix of 369 × k and used a notation of 1-0-0 if the first data point belonged to Cluster 1. Conversely, Researcher B may have created a vector of 369 × 1 and written 1 (or 2) as the vector element for the first data point if it belonged to Cluster 1 (or Cluster 2);

2. Computing mean values: After the parameters were specified, a mean value was calculated per cluster. In the Solution Display step, a mean value was computed for each attribute based only on the data points in Cluster 1. Additionally, attribute means were computed by using only the data points in Clusters 2–k;

3. Computing the distances between the mean and the data points: The Euclidean distances between the empirical data points and the attribute mean (calculated in Step 2) were computed. For Clusters 1–k, the Euclidean distances between the mean and data points were summed to obtain k Euclidean distance indices;

4. Examining final indices: A final index was computed by adding the k indices generated in Step 3. For additional final indices, the steps above were iterated by using a different solution display in Step 1. At this point, a data point in Cluster 1 may be moved to Cluster 2 or to a randomly selected cluster. Once a different solution display is created, Steps 2–4 should be repeated to obtain a better index.

In the study, the most popular algorithm, K-means, was used to ensure the reliability, stability, and generalizability of the study findings. Step 5 was added to overcome limitations of the evaluation method using Euclidean distances.

5.  Finalizing evaluations: A silhouette analysis was performed on the data according to the final solution display for which the exploration was completed. Next, the number of clusters, k, was increased from 2 to 16 to compute the silhouette coefficients, and the solution with the largest coefficient value was determined as the final product.

Silhouette analysis was developed by Rousseeuw in 1987 [13]. As the number of clusters, k, increases, the Euclidean distance index shows a decreasing trend because the number of data points per cluster steadily reduces. Moreover, with Euclidean distances, only relative evaluation is possible. Accordingly, clustering solutions can be compared based on Euclidian distances to determine the best result only if the number of clusters, k, is the same. Rousseeuw's [13] silhouette function measures relative distances among the data points in a cluster, and the ratio between maximum and minimum distances is used to determine the best solution.

For each data point x in the data set, there is $S(x)$, which is the silhouette coefficient between the data point x and another data point apart from x in the data set. Based on clusters A, B, and C, $a(x)$ is the average distance between data point $x$ and another data point except for $x$ in cluster A and the average distance between $a(x)$ and each data point in clusters B and C. d(x, B) and d(x, C) are the average distances between $a(x)$ and each data point included in clusters B and C, respectively. If d(x, B) is smaller than d(x, C), $b(x)$ is set d(x, B). When these conditions are satisfied, the silhouette coefficient for the data point $x$ can be obtained using the following Equation (2), and the final silhouette coefficient for all data can be obtained through the Equation (3).

$$S(x) = \{b(x) - a(x)\}/\max\{a(x),\ b(x)\} \tag{2}$$

$$Maximize\ \left(\frac{1}{n}\right) \sum_{i=1}^{n} S(x_i) \tag{3}$$

Accordingly, as the distances between data points in a cluster decrease and that between the data points and the cluster closest to the first cluster increase, silhouette coefficients improve. The range of the silhouette coefficient is $-1$ to 1. The closer to 1 the silhouette coefficient is, the better the clustering solution, because data points within a cluster are densely grouped, and data points between different clusters are far apart.

The K-means algorithm was chosen because it has been used for the longest time and is stable. Additionally, it is possible to ensure reliability and generalizability, as shown in the literature. It has also been used in some previous studies on gamification [14,15]. According to Knutas et al. [14], the K-means algorithm was used to derive the learner's preferred profile in a gamified learning environment. Knutas et al. derived four types of learner profiles and explained them using Bartle's four player types. Marisa et al. [15] analyzed the responses of 63 learners using the K-means algorithm to derive the gamification factors necessary for collaboration in a gamified learning environment. They also derived decision-based results using the fuzzy analytical hierarchy process technique.

However, the K-means algorithm has certain limitations. The primary limitation is a local solution, which is not the final choice amongst one of the many solutions found before reaching the optimal one. K-means does not find the optimal solution unless a separate algorithm or option is added. There is no additional feature on the initial solution setting. Additionally, K-means faces challenges in manual data outliers and clustering parameter settings, such as, number of cluster settings [16].

To solve problems with a local solution of a clustering algorithm, many solutions have been derived. Selim and Alsultan [17] developed the Simulated Annealing (SA) algorithm-based clustering. The SA algorithm is based on the process that mimics the quenching process of refining steel. The SA algorithm is powerful but requires a long time to calculate the data clustering. The reason for this is that the SA algorithm accepts probabilistically bad results, and finds the optimal clustering over a wider range, unlike other algorithms.

Pal and Saraswat [18] developed the Biogeography Based Optimization (BBO) algorithm, which includes biogeography elements such as habitat, mutation, immigration, and

emigration. The BBO algorithm clusters, using the immigration and emigration rate-based algorithm. In the solution setting process, the immigration rate and emigration rate are analyzed. According to the analyzed rates, each datapoint that raises the evaluation result of the cluster is collected, along with the collection of data that lower the evaluation result of the cluster. Through this process, the BBO algorithm finds a better cluster. It is also possible for the BBO algorithm to solve the local solution problem, but with a calculation time much longer than K-means.

## 3. Results

Data analysis was performed using R studio 2022.07.1 Build 554 version loaded in a PC notebook outfitted with a 12th Gen Intel® Core™ I5-1240P 1.70 GHz CPU, 16 GB of RAM, and an Nvidia GeForce RTX 2050 graphics card.

To perform K-means and silhouette analysis, the "Cluster" package was used; to visualize clustering results, "doBy" and "fmsb"' packages were used. In addition, when the K-means algorithm was executed, it was iterated 1000 times each and the highest silhouette coefficient was set as the final result. The doBy and fmsb packages include commands to visualize three or more variables on 2D graphs. In general, three or more variables are visualized using 3D graphs, but they are inferior to 2D graphs with respect to readability. This study evaluated four attributes, and "radar charts" were deemed the most effective means for visualizing the clustering results in 2D graphs. Hence, the two aforementioned packages were used.

Table 2 shows the results of descriptive statistics conducted on raw and standardized data (369 data points). The data were collected from the college courses taught by members of the current research team between 2016 and 2019 in which PBL and game results were considered to grade academic performance. Student demographic information, such as school year or gender, were not included in the dataset. Additionally, the data analyzed in the study were collected from students from several colleges rather than only the institution with which the research team was affiliated. Table 3 shows the results of K-means clustering performed on 369 data points and four attributes and silhouette analysis performed on the displays derived from the K-means. The number of clusters, k, varied from 2 to 16, which was determined by considering the minimal number of clusters (k = 2) and the total number of MBTI personality types (k = 16) based on silhouette coefficient results.

**Table 2.** Description analysis results.

| Lecture | Point | Average | Min | Max | Average (Normal) | Min (Normal) | Max (Normal) |
|---|---|---|---|---|---|---|---|
| 2016/2nd/ Gamification Engineering | Activity | 29.069 | 21 | 30 | 0.897 | 0 | 1 |
| | Game | 33.332 | 28 | 40.4 | 0.43 | | |
| | Project | 45.943 | 40 | 51.5 | 0.517 | | |
| | Exam | 44.276 | 20 | 58 | 0.639 | | |
| 2017/1st/Entrepreneurship | Activity | 76.633 | 64 | 92 | 0.451 | 0 | 1 |
| | Game | 110.133 | 72 | 142 | 0.545 | | |
| | Project | 21.05 | 18.4 | 23.8 | 0.491 | | |
| | Exam | 19.667 | 14 | 30 | 0.354 | | |
| 2017/2nd/Gamification Engineering | Activity | 44.862 | 28 | 66 | 0.444 | 0 | 1 |
| | Game | 13.793 | 8 | 20 | 0.483 | | |
| | Project | 20.172 | 15 | 25 | 0.517 | | |
| | Exam | 13.983 | 3 | 19 | 0.686 | | |
| 2017/1st/Technology Innovation | Activity | 52.606 | 43 | 64 | 0.548 | 0 | 1 |
| | Game | 23.506 | 9.7 | 24.9 | 0.908 | | |
| | Project | 16.878 | 0 | 19 | 0.888 | | |
| | Exam | 23.091 | 7.5 | 29 | 0.725 | | |
| 2018/1st/Technology Innovation | Activity | 61.6 | 53 | 77 | 0.43 | 0 | 1 |
| | Game | 45.767 | 32 | 57 | 0.551 | | |
| | Project | 22.472 | 19.41 | 25 | 0.548 | | |
| | Exam | 30.333 | 18 | 37.5 | 0.632 | | |

**Table 3.** Silhouette coefficient results.

| K | Silhouette Coefficient |
|---|---|
| 2 | 0.3167 |
| 3 | **0.3506** |
| 4 | 0.2998 |
| 5 | 0.2375 |
| 6 | 0.2226 |
| 7 | 0.236 |
| 8 | 0.2499 |
| 9 | 0.2511 |
| 10 | 0.2028 |
| 11 | 0.2536 |
| 12 | 0.2633 |
| 13 | 0.2327 |
| 14 | 0.2303 |
| 15 | 0.2197 |
| 16 | 0.2343 |
| Average | 0.25339 |
| Min | 0.2028 |
| Max | 0.3506 |

The silhouette coefficient was highest at 0.3506 when k = 3 and lowest at 0.2028 when k = 10 (Figure 4).

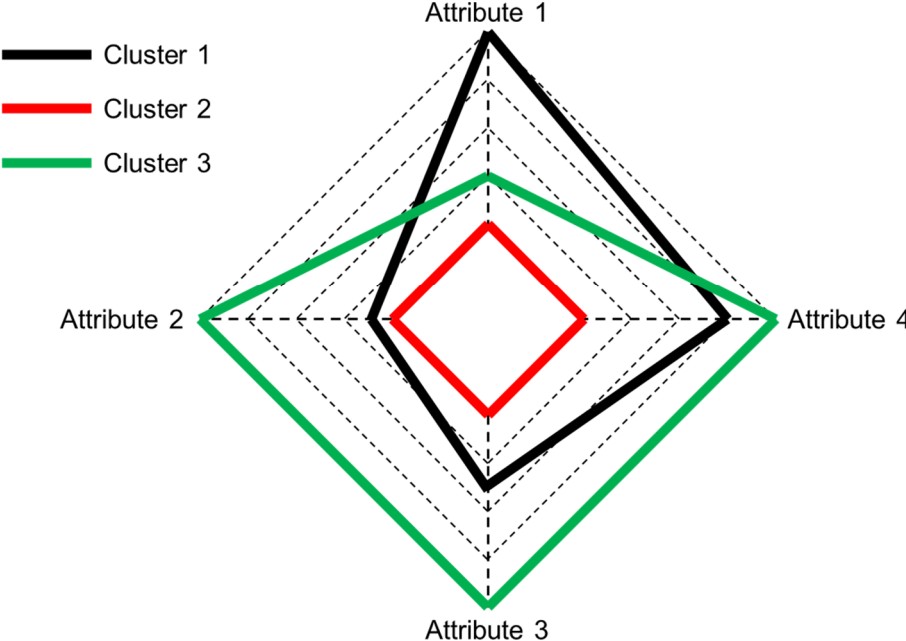

**Figure 4.** Radar chart based on K means analysis. Black, red, and green lines show the mean values in Clusters 1, 2, and 3, respectively.

Cluster 1 had the highest mean value for Activity Points and was second highest for the other three attributes. Cluster 2 showed the lowest mean value across all of the attributes. Cluster 3 had the highest value among the three clusters for all attributes except for Activity Points.

## 4. Discussion

Three learning performance patterns were derived across several classes taught using gamification strategies. The radar chart visualizing cluster means confirms that the clusters showed distinct characteristics. Evaluation of the data shows that Cluster 1 comprised a group whose focus was earning Activity and Exam Points, Cluster 2 showed low performance across the attributes, and Cluster 3 showed high performance across three attributes, excluding Activity Points.

To explain the differences in the characteristics across clusters, we referred to self-regulated learning and achievement goal-orientation theories. Pintrich [19] defined self-regulated learning as a learning process in which learners set goals, develop strategies, actively utilize learning resources, and self-evaluate their learning outcomes to provide feedback as well as promote growth. A prerequisite for self-regulated learning is active participation in learning environments; once a learner devises learning goals, develops strategies, and sets objectives, they can self-regulate their learning [20]. Gamified learning environments induce self-regulated learning based on the game mechanics and rules [21]. Thus, learners in a gamified class self-regulate learning, which occurs based on game mechanics and game rules specified by the instructor and interactions with other learners.

In addition, the achievement goal-orientation theory operates through the process of goal setting. Kim et al. [2] stated that during gamification in education, learners participate in learning by setting mastery, performance, and avoidance goals as defined in the achievement goal-orientation theory. Mastery goals refer to goals set by learners focusing on the learning activity itself, such as mastering a task or improving one's understanding. According to Kaplan et al. [22], performance goals focus on comparing a learner's competence against that of others. Avoidance goals focus on the learner's need to avoid or hide their own incompetence and avoid negative feedback.

In a gamified class, learners rely on self-regulated learning to identify usable learning resources and establish study strategies. As learners set goals based on their preferred strategies, they may weigh mastery, performance, and avoidance goals differently.

We interpreted the cluster characteristics as follows: In Cluster 1, the performance goal regarding Activity Points was the most dominant; and in Cluster 3, the performance goal regarding attributes other than Activity Points was the most dominant. Compared with the other point categories, Activity Points provided feedback most immediately because the learners could determine their earned points through the leaderboard. Hence, we contend that Cluster 1 learners emphasized performance goals. In categories other than Activity Points, feedback was not immediate, and the learners discovered their performance level only as the semester progressed. Thus, learners in Cluster 3 participated in learning primarily by setting mastery goals to improve their competence rather than getting ahead of other learners.

Cluster 2 learners did not appear to exhibit self-regulated learning for the following three reasons: First, we speculate that they had low levels of self-control. According to Schunk and Zimmerman [23], learners with low self-control have difficulty engaging in self-regulated learning and are thus likely to experience difficulties in utilizing learning resources and developing strategies. Consequently, they did not actively participate in learning activities. Second, it is possible that their perceived failure could have been amplified after continued poor performance. Downward counterfactual thinking refers to the mode of thinking characterized by the phrase "It is of no use, even if I keep trying" due to continuous experiences of failure [24]. In a gamified class, learners may amass failure experiences because of a series of failures and give up on learning based on downward counterfactual thinking [25]. Third, there may have been mismatches between mission difficulty and student level. According to Nakamura et al. [26], if the task difficulty does not match with the user level, the users feel burdened by the task and may easily give up. Learners may give up on learning because their ability does not correspond to the difficulty level of the mission or game. These three reasons suggest that learners in Cluster 2 could not experience self-regulated learning or adjust to the gamified class.

This study uses the K-means algorithm for data clustering since this algorithm is verified and validated. Additionally, this study used K-means because of strong stability against data noise or outsiders [27]. The K-means algorithm has been used for a long time and has derived numerous algorithms during that period. The advantage of K-means is its calculation time, where this algorithm works faster than other clustering algorithms, such as fuzzy c-means [28].

## 5. Conclusions

This study aimed to identify key elements that may influence the effectiveness of gameful experience design in sustainable education environments. In the study, gamified college classes were designed by applying two types of gamifications: the PBL system based on points, badges, and a leaderboard, which allowed learners to develop their own strategies in class; and educational games, which were developed for learners to engage with course content through play environments. These games were designed to enable learners to apply the topics they learned in class while playing a board game or to evaluate other learners' projects in a gamified format.

Based on these gamification types, a grade point system was designed to include Activity, Game, Project, and Exam Points. These categories were treated as attributes during data analysis. A total of 369 data points were collected across five college courses taught by the research team between 2016 and 2019. To identify learning performance patterns, a K-means data clustering algorithm was used; to determine the optimal solution, the clustering results were evaluated by computing silhouette coefficients.

Learning performance clusters were derived from all students' performance data. The patterns were interpreted by referring to self-regulated learning and achievement goal-orientation theories. Students in Cluster 1 appeared to emphasize performance goals, whereas those in Cluster 3 focused on mastery goals. Students in Cluster 2 did not appear to emphasize any self-guided goal strategies and may have been characterized by poor self-control, downward counterfactual thinking due to accumulated failure experience, and a mismatch between mission difficulty and learner level.

The study findings are expected to make a useful contribution to the literature on gamification in educational environments by facilitating researchers to determine the optimum game difficulty level and mission setting for designing sustainable gamified classes. In designing a PBL system-based gamified class, mission difficulty and quantity should be controlled because different learners have different attitudes and objectives for taking the course. Accordingly, with respect to the mission difficulty, a mission should be determined by considering the three academic performance patterns identified in this study. In addition, the differential learning patterns identified in the study should be considered when determining the main mission, which is directly linked to class content to support student learning.

Moreover, our findings support levels and gameful experience design. Typically, level and experience points are used together for learners to earn experience points by accomplishing a mission and move to a higher level once they earn points above a certain threshold. In designing a stimulator to raise learners to a higher level, considering differential learning patterns is expected to improve the application and effectiveness of level and experience design.

Despite the contributions of this study, it has the following limitations: absence of student interviews and data for online gamified classes. The study findings were derived based on empirical data. Thus, it is expected that if interviews with students in each cluster were conducted and the results were incorporated with our findings, its reliability and generalizability would improve. In future research, focus group interviews with students in each cluster should be arranged to improve the comprehensiveness of the findings. Moreover, this study was conducted only with the data obtained prior to COVID-19. It was reasoned that COVID-19 provided a unique learning situation and online learning during the pandemic would thus be different from previous online classes. Future studies could

compare online gamified class data between COVID-19 and non-COVID-19 situations and integrate data from online and offline gamified classes to facilitate the design of specific gameful experiences for complex educational contexts encountered by instructors.

**Author Contributions:** Conceptualization, S.P.; formal analysis, S.P.; funding acquisition, S.K.; methodology, S.P.; writing—original draft, S.P.; writing—review and editing, S.K. All authors have read and agreed to the published version of the manuscript.

**Funding:** This research was supported by a grant from the National Research Foundation of Korea and funded by the Korean Government (Ministry of Science and ICT; #2020R1A2B501001801). The research received support from the MSIT (Ministry of Science and ICT), Korea, under the ITRC (Information Technology Research Center) support program (IITP-2021-0-02051) supervised by the IITP (Institute for Information & Communications Technology Planning & Evaluation).

**Institutional Review Board Statement:** Not applicable.

**Informed Consent Statement:** Not applicable.

**Data Availability Statement:** Not applicable.

**Conflicts of Interest:** The authors declare no conflict of interest.

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
