# Peer review of "Learning Performance Styles in Gamified College Classes Using Data Clustering"

_sustainability, doi:10.3390/su142315574_

Round 1

Reviewer 1 Report

Overall, I think that this work presents an interesting idea (Application of a clustering technique for educational purposes). However, I have many concerns about the clustering technique used. I describe my general and specific comments below

General comments:

  1. Authors describe the use of a single clustering algorithm. However, it is very common to employ different clustering algorithms and combine those outputs to improve the robustness of clustering (This is because the individual behaviors of the different algorithms are exploited by joining performances). Have the authors considered the use of clustering aggregation techniques?

  1. In addition, as authors suggest, K-Means is an efficient and well-known algorithm. However, it also involves some weaknesses that are not presented in the manuscript (e.g. It was designed to identify homogeneous spherically-shaped clusters). I think the choice of this specific algorithm should also be better justified. Have the authors considered at least other methods such as Fuzzy-C-Means or Hierarchical clustering?

  1. K-Means algorithm requires a number of clusters as input. Authors describe the use of Silhouette coefficients to choose the optimal number of groups. However, numerous metrics have been proposed over time with similar purposes (E.g. Partition coefficient, entropy coefficient, partition index, separation index…). I think authors should better justify this selection.

  1. Authors describe that the Silhouette coefficient ranges between -1 to 1. The closer to 1, the better the clustering solution. However, the best partition here involves a coefficient equal to 0.35. It seems a value too far from unity and I think a deeper discussion about these low values ​​is needed. Can it really be affirmed that the partition is optimal with such low values?

Specific comments:

  1. It seems that Figure 1,2,3 and 4 are low quality images. I recommend increasing quality by means of a higher dpi.

  1. Line 244. Previous relevant studies should be described in detail in the introduction.

  1. Line 233. Relevant mathematical functions and equations should be formally included in the document (or at least in an appendix or supplementary material).

  1. Section 2.3. I think a flowchart would improve the clarity of these explanations.

Reviewer 2 Report

This topic is important nowadays. The treatment is correct but results can be compared with other in general context. There are a lot of experiences similar in educative context and this can be showed. 

Reviewer 3 Report

The objective of the article is to investigate the performance of gamified learning trying to answer the following research question: How many patterns of learning performance in a gamified class can be identified and what are the characteristics of each one?

The manuscript should clearly differentiate two sections: Introduction and Theoretical Framework. In addition, the second section allows us to deepen and reinforce the idea of gamification as a teaching-learning tool by introducing more quotes or experiences with this methodology in higher education or other levels. This section will help the reader understand all the aspects that the authors want to cover and better detect the indicators of strengths and weaknesses of gamification integration in the teaching-learning process, both in pandemic situations and at other times. For example, these two quotes are recommended from which some other reference can be obtained:

DOI: https://doi.org/10.6018/red/60/05

DOI: https://doi.org/10.6018/red.439981

Regarding the method, it is well justified globally, however, the sample is not made clear. In this sense, when the authors refer to 369 points, are they referring to 369 students belonging to the subjects they comment on? On the other hand, it is not clearly specified to which studies the subjects in which gamification is applied belong. is it degree in multimedia resources or similar?

For the subjects mentioned (closely related to gamification) possibly gamified learning is adequate to address the contents, however, would it be possible to analyze a large part of the contents of other subjects not so closely related to gamification? , in this sense, the authors want to extend this idea to all higher education and perhaps they should be more cautious with the generalization. However, although in the conclusions the authors comment that in order to generalize it is necessary to conduct interviews with the students, they do not make any reference to the generalization for all subjects.

What do the authors mean when they refer to a “sustainable education environment” in the text? what is available to everyone?

As formal aspects, it is allowed to cite the figures after inserting them in the text, but it is necessary to always cite them before they appear in the text. This occurs with figure 2 which is cited later (line 84), however, it has not been presented before being inserted in the text. The same thing happens with table 2 and with figure 4. Please, you should review this question about figures and tables throughout the text. On the other hand, the criteria of upper or lower case should be unified when writing Activity Points or activity points.

Round 2

Reviewer 1 Report

I think the authors have vaguely considered my previous comments:

1.The last comment stated: "Section 2.3 I think a flowchart would improve the clarity of these explanations". This comment was ignored in the author's response.

2. As pointed out in the previous review, I think the authors should better justify the selection of silhoutte coefficients. In their answer, they vaguely comment on some difference with Euclidean distance based metrics. However, this answer is not added to the manuscript by means of additional sentences. I strongly recommend to clearly state in the manuscript the advantage of Silhoute coefficients against at least the partition coefficient or the entropy coefficient.

3. Authors discuss that the expected values for Silhouete coefficients should range between 0.4 and 0.6. Please, add those explanations and corresponding references to the manuscript. Why the value obtained here is lesser than 0.4?

4. I strongly recommend to clearly state advantages and limitations of k-Means against other clustering techniques (at least Fuzzy-C-Means and hierarchical clustering).

Round 3

Reviewer 1 Report

I insist that the authors should include in the manuscript a discussion of the advantages and disadvantages of the k-means method compared to other clustering techniques. I also insist that a discussion of the quality metric used compared to other alternatives should be included in the document.

Round 4

Reviewer 1 Report

Authors have finally taken my comments into consideration. They have included a better description about of the K-Means algorithm and its limitations, which was my main concern.